# ADVERSARIAL ROBUSTNESS AS A PRIOR FOR LEARNED REPRESENTATIONS

## ABSTRACT

An important goal in deep learning is to learn versatile, high-level *feature representations* of input data. However, standard networks' representations seem to possess shortcomings that, as we illustrate, prevent them from fully realizing this goal. In this work, we show that *robust optimization* can be re-cast as a tool for enforcing *priors* on the features learned by deep neural networks. It turns out that representations learned by robust models address the aforementioned shortcomings and make significant progress towards learning a high-level encoding of inputs. In particular, these representations are approximately invertible, while allowing for direct visualization and manipulation of salient input features. More broadly, our results indicate adversarial robustness as a promising avenue for improving learned representations. [1]

## 1 INTRODUCTION

Beyond achieving remarkably high accuracy on a variety of tasks (Krizhevsky et al., 2012; He et al., 2015; Collobert & Weston, 2008), a major appeal of deep learning is the ability to learn effective *feature representations* of data. Specifically, deep neural networks can be thought of as linear classifiers acting on *learned feature representations* (also known as *feature embeddings*). A major goal in representation learning is for these embeddings to encode high-level, interpretable features of any given input (Goodfellow et al., 2016; Bengio et al., 2013; Bengio, 2019). Indeed, learned representations turn out to be quite versatile—in computer vision, for example, they are the driving force behind transfer learning Girshick et al. (2014); Donahue et al. (2014), and image similarity metrics such as VGG distance Dosovitskiy & Brox (2016a); Johnson et al. (2016); Zhang et al. (2018).

These successes and others clearly illustrate the utility of learned feature representations. Still, deep networks and their embeddings exhibit some shortcomings that are at odds with our idealized model of a linear classifier on top of interpretable high-level features. For example, the existence of adversarial examples (Biggio et al., 2013; Szegedy et al., 2014)—and the fact that they may correspond to flipping predictive features Ilyas et al. (2019)—suggests that deep neural networks make predictions based on features that are vastly different from what humans use, or even recognize. (This message has been also corroborated by several recent works (Brendel & Bethge, 2019; Geirhos et al., 2019; Jetley et al., 2018; Zhang & Zhu, 2019).) In fact, we show a more direct example of such a shortcoming (c.f. Section 2), wherein one can construct pairs of images that appear completely different to a human but are nearly identical in terms of their learned feature representations.

**Our contributions.** Motivated by the limitations of standard representations, we propose using the robust optimization framework as a tool to enforce (user-specified) *priors* on features that models should learn (and thus on their learned feature representations). We demonstrate that the resulting learned "robust representations" (the embeddings learned by adversarially robust neural networks Goodfellow et al. (2015); Madry et al. (2018)) address many of the shortcomings affecting standard learned representations and thereby enable new modes of interaction with inputs via manipulation of salient features. These findings are summarized below (c.f. Figure 1 for an illustration):

---

[1]Our code and models for reproducing these results is available at https://github.com/snappymanatee/robust-learned-representations

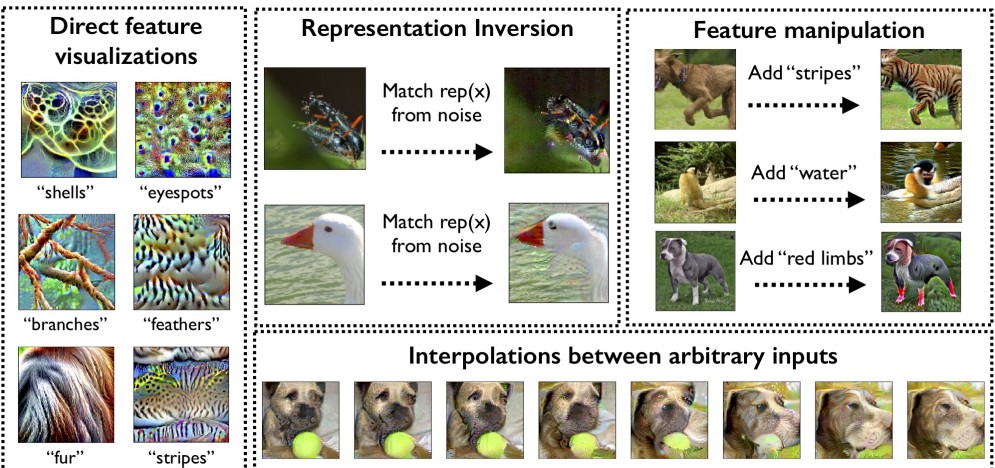

Figure 1: Sample images highlighting the properties and applications of "robust representations" studied in this work. All of these manipulations use only gradient descent on simple, unregularized, direct functions of the representations of adversarially robust neural networks Goodfellow et al. (2015); Madry et al. (2018).

- **Representation inversion (Section 4.1)**: In stark contrast to standard representations, robust representations are *approximately invertible*—that is, they provide a high-level embedding of the input such that images with similar robust representations are semantically similar, and the salient features of an image are easily recoverable from its robust feature representation. This property also naturally enables feature interpolation between arbitrary inputs.

- **Simple feature visualization (Section 4.2)**: Direct maximization of the coordinates of robust representations suffices to visualize easily recognizable features of the model. This is again a significant departure from standard models where (a) without explicit regularization at visualization time, feature visualization often produces unintelligible results; and (b) even with regularization, visualized features in the representation layer are scarcely human-recognizeable Olah et al. (2017).

- **Feature manipulation (Section 4.2.1)**: Through the aforementioned direct feature visualization property, robust representations enable the addition of specific features to images through direct first-order optimization.

Broadly, our results indicate that robust optimization is a promising avenue for learning representations that are more "aligned" with our notion of perception. Furthermore, our findings highlight the the desirability of adversarial robustness as a goal beyond the standard security and reliability context.

## 2 LIMITATIONS OF STANDARD REPRESENTATIONS

Following standard convention, for a given deep network we define the *representation* $R(x) \in \mathbb{R}^k$ of a given input $x \in \mathbb{R}^d$ as the activations of the penultimate layer of the network (where usually $k \ll d$). The prediction of the network can thus be viewed as the output of a linear classifier on the representation $R(x)$. We refer to the *distance in representation space* between two inputs $(x_1, x_2)$ as the $\ell_2$ distance between their representations $(R(x_1), R(x_2))$, i.e., $\|R(x_1) - R(x_2)\|_2$.

A common aspiration in representation learning is to have that for any pixel-space input $x$, $R(x)$ is a vector encoding a set of "human-meaningful" features of $x$ Bengio (2019); Goodfellow et al. (2016); Bengio et al. (2013). These high-level features would be linearly separable with respect to the classification task, allowing the classifier to attain high accuracy.

Running somewhat counter to this intuition, however, we find that it is straightforward to construct pairs of images with nearly identical representations yet drastically different content, as shown in

Figure 2. Finding such pairs turns out to be as simple as sampling two images $x_1, x_2 \sim \mathcal{D}$, then optimizing one of them to minimize distance in representation space to the other:

$$x_1' = x_1 + \arg\min_\delta \|R(x_1 + \delta) - R(x_2)\|_2. \tag{1}$$

Indeed, solving objective (1) yields images that have similar representations, but share no qualitative resemblance (in fact, $x_1'$ tends to look nearly identical to $x_1$). An example of such a pair is given in Figure 2.

Note that if representations truly provided an encoding of any image into high-level features, finding images with similar representations should necessitate finding images with similar high-level features. Thus, the existence of these image pairs (and similar phenomena observed by prior work Jacobsen et al. (2019)) lays bare a misalignment between the notion of distance induced via the features learned by current deep networks, and the notion of distance as perceived by humans.

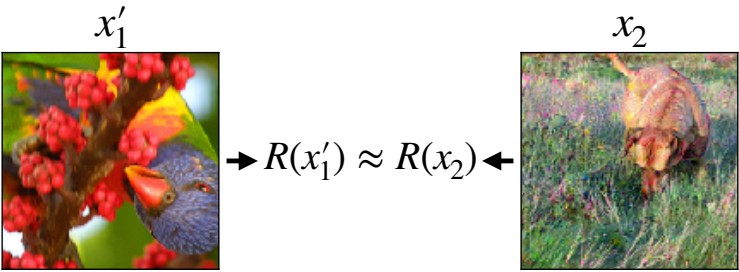

Figure 2: A limitation of standard neural network representations: it is straightforward to construct pairs of images $(x_1', x_2)$ that appear completely different yet map to similar representations.

## 3   ADVERSARIAL ROBUSTNESS AS A PRIOR

Our analysis in Section 2 and prior work (Jacobsen et al., 2019) prompt the question:

*How can we learn better-behaved representations?*

In this work, we demonstrate that the representations learned by adversarially robust neural networks seem to address many identified limitations of standard representations, and make significant progress towards the broader goal of learning high-level, human-understandable encodings.

**Adversarially robust deep networks and robust optimization.** In standard settings, supervised machine learning models are trained by minimizing the expected loss with respect to a set of parameters $\theta$, i.e., by solving an optimization problem of the form:

$$\theta^* = \min_\theta \mathbb{E}_{(x,y)\sim\mathcal{D}} \left[ \mathcal{L}_\theta(x, y) \right]. \tag{2}$$

We refer to (2) as the *standard* training objective—finding the optimum of this objective should guarantee high performance on unseen data from the distribution. It turns out, however, that deep neural networks trained with this standard objective are extraordinarily vulnerable to *adversarial examples* (Biggio et al., 2013; Szegedy et al., 2014)—by changing a natural input imperceptibly, one can easily manipulate the predictions of a deep network to be arbitrarily incorrect.

A natural approach (and one of the most successful) for defending against these adversarial examples is to use the *robust optimization framework*: a classical framework for optimization in the presence of uncertainty (Wald, 1945; Danskin, 1967). In particular, instead of just finding parameters which minimize the expected loss (as in the standard objective), a robust optimization objective also requires that the model induced by the parameters $\theta$ be robust to worst-case perturbation of the input:

$$\theta^* = \arg\min_\theta \mathbb{E}_{(x,y)\sim\mathcal{D}} \left[ \max_{\delta\in\Delta} \mathcal{L}_\theta(x + \delta, y) \right]. \tag{3}$$

This robust objective is in fact common in the context of machine learning security, where $\Delta$ is usually chosen to be a simple convex set, e.g., an $\ell_p$-ball. Canonical instantiations of robust optimization such as adversarial training (Goodfellow et al., 2015; Madry et al., 2018)) have arisen as practical ways of obtaining networks that are invariant to small $\ell_p$-bounded changes in the input while maintaining high accuracy (though a small tradeoff between robustness and accuracy has been noted by prior work Tsipras et al. (2019); Su et al. (2018)(also cf. Appendix Tables 4 and 5 for a comparison of accuracies of standard and robust classifiers)).

**Robust optimization as a feature prior.** Traditionally, adversarial robustness in the deep learning setting has been explored as a goal predominantly in the context of ML security and reliability (Biggio & Roli, 2018).

In this work, we consider an alternative perspective on adversarial robustness—we cast it as a prior on the features that can be learned by a model. Specifically, models trained with objective (3) must be *invariant* to a set of perturbations $\Delta$. Thus, selecting $\Delta$ to be a set of perturbations that humans are robust to (e.g., small $\ell_p$-norm perturbations) results in models that share more invariances with (and thus are encouraged to use similar features to) human perception. Note that incorporating human-selected priors and invariances in this fashion has a long history in the design of ML models—convolutional layers, for instance, were introduced as a means of introducing an invariance to translations of the input (Fukushima, 1980).

In what follows, we will explore the effect of the prior induced by adversarial robustness on models' learned representations, and demonstrate that representations learned by adversarially robust models are better behaved, and do in fact seem to use features that are more human-understandable.

## 4 PROPERTIES AND APPLICATIONS OF ROBUST REPRESENTATIONS

In the previous section, we proposed using *robust optimization* as a way of enforcing user-specified priors during model training. Our goal was to mitigate the issues with standard representations identified in Section 2. We now demonstrate that the learned representations resulting from training with this prior indeed exhibit several advantages over standard representations.

Recall that we define a representation $R(\cdot)$ as a function induced by a neural network which maps inputs $x \in \mathbb{R}^n$ to vectors $R(x) \in \mathbb{R}^k$ in the representation layer of that network (the penultimate layer). In what follows, we refer to "standard representations" as the representation functions induced by standard (non-robust) networks, trained with the objective (2)—analogously, "robust representations" refer to the representation functions induced by $\ell_2$-adversarially robust networks, i.e. networks trained with the objective (3) with $\Delta$ being the $\ell_2$ ball:

$$\theta^*_{robust} = \arg\min_\theta \mathbb{E}_{(x,y)\sim\mathcal{D}} \left[ \max_{\|\delta\|_2 \le \varepsilon} \mathcal{L}_\theta(x + \delta, y) \right].$$

It is worth noting that despite the value of $\varepsilon$ used for training being quite small, we find that robust optimization *globally* affects the behavior of learned representations. As we demonstrate in this section, the benefits of robust representations extend to out-of-distribution inputs and far beyond $\varepsilon$-balls around the training distribution.

**Experimental setup.** We train robust and standard ResNet-50 (He et al., 2016) networks on the Restricted ImageNet (Tsipras et al., 2019) and ImageNet (Russakovsky et al., 2015) datasets. Datasets specifics are in in Appendix A.1, training details are in in Appendices A.2 and A.3, and the performance of each model is reported in Appendix A.4. In the main text, we present results for Restricted ImageNet, and link to (nearly identical) results for ImageNet present in the appendices (B.1.4,B.3.2).

Unless explicitly noted otherwise, our optimization method of choice for any objective function will be (projected) gradient descent (PGD), a first-order method which is known to be highly effective for minimizing neural network-based loss functions for both standard and adversarially robust neural networks (Athalye et al., 2018a; Madry et al., 2018).

Code for reproducing our results is available at `https://github.com/snappymanatee/robust-learned-representations`.

### 4.1 INVERTING ROBUST REPRESENTATIONS

As discussed in Section 2, for standard deep networks, given any input $x$, it is straightforward to find another input that looks entirely different but has nearly the same representation (c.f. Figure 2). We noted that this finding runs somewhat counter to the idea that these learned representations effectively capture relevant input features. After all, if the representation function was truly extracting "high-level" features of the input as we conceptualize them, semantically dissimilar images should (by definition) have different representations. We now show that the state of affairs is greatly improved for robust representations.

**Robust representations are (approximately) invertible out of the box.** We begin by recalling the optimization objective (1) used in Section 2 to find pairs of images with similar representations, a simple minimization of $\ell_2$ distance in representation space from a source image $x_1$ to a target image $x_2$:

$$x'_1 = x_1 + \arg\min_{\delta} \|R(x_1 + \delta) - R(x_2)\|_2. \tag{4}$$

This process can be seen as recovering an image that maps to the desired target representation, and hence is commonly referred to as *representation inversion* (Dosovitskiy & Brox, 2016b; Mahendran & Vedaldi, 2015; Ulyanov et al., 2017). It turns out that in sharp contrast to what we observe for standard models, the images resulting from minimizing (4) for robust models are actually *semantically similar* to the original (target) images whose representation is being matched, and this behavior is consistent across multiple samplings of the starting point (source image) $x_1$ (cf. Figure 3).

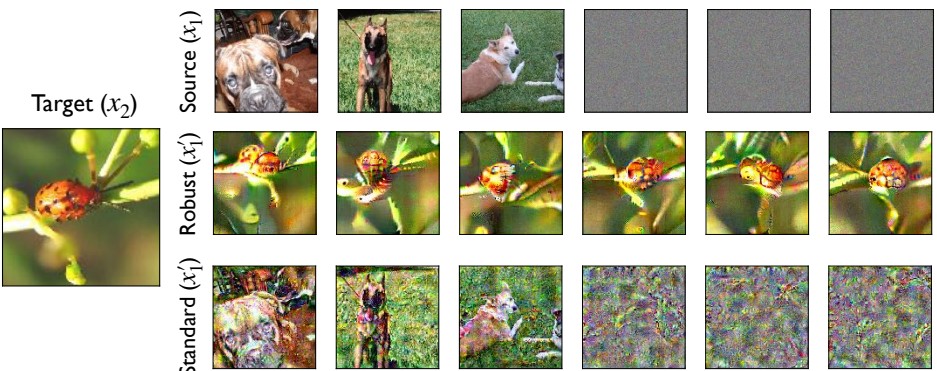

Figure 3: Visualization of inputs that are mapped to similar representations by models trained on the Restricted ImageNet dataset. *Target ($x_2$) & Source ($x_1$)*: random examples image from the test set; *Robust* and *Standard ($x'_1$)*: result of minimizing the objective (4) to match (in $\ell_2$-distance) the representation of the target image starting from the corresponding source image for (*top*): a robust (adversarially trained) and (*bottom*): a standard model respectively. For the robust model, we observe that the resulting images are perceptually similar to the target image in terms of high-level features (even though they do not match it exactly), while for the standard model they often look more similar to the source image which is the seed for the optimization process. Additional results in Appendix B.1, and similar results for ImageNet are in Appendix B.1.4.

**Representation proximity seems to entail semantic similarity.** In fact, the contrast between the invertibility of standard and robust representations is even stronger. To illustrate this, we will attempt to match the representation of a target image while staying close to the starting image of the optimization in pixel-wise $\ell_2$-norm (this is equivalent to putting a norm bound on $\delta$ in objective (4)). With standard models, we can consistently get close to the target image in representation space, without moving far from the source image $x_1$. On the other hand, for robust models, we cannot get close to the target representation while staying close to the source image—this is illustrated quantitatively in Figure 4. This indicates that for robust models, semantic similarity may in fact be necessary for representation similarity (and is not, for instance, merely an artifact of the local robustness induced by robust optimization).

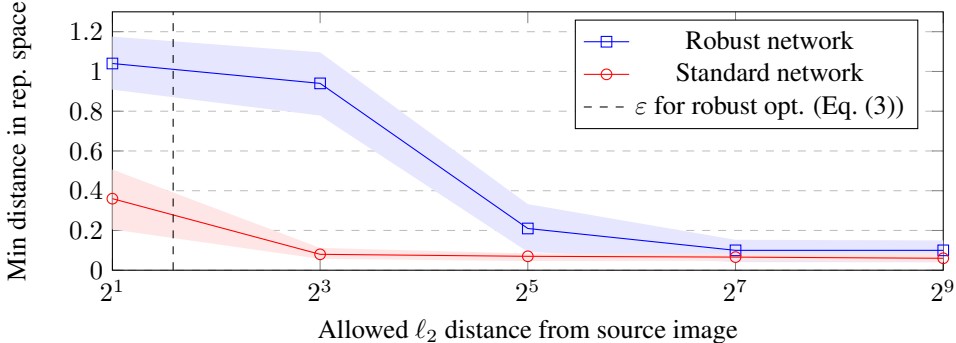

Figure 4: Optimizing objective (4) with PGD and an $\ell_2$-norm constraint around the source image. On the $x$-axis is the radius of the constraint set, and on the $y$-axis is the distance in representation space between the minimizer of objective (4) within the constraint set and the target image, normalized by the norm of the representation of the target image: i.e., a point $(x_i, y_i)$ on the graph corresponds to $y_i = \min_{\|\delta\|_2 \leq x_i} \|R(x + \delta) - R(x_{targ})\|_2 / \|R(x_{targ})\|_2$. Notably, we are unable to closely match the representation of the target image for the robust network until the norm constraint grows very large, and in particular much larger than the norm of the perturbation that the model is trained to be robust against ($\varepsilon$ in objective (3)). Shown are 95% confidence intervals over random choice of source and target images.

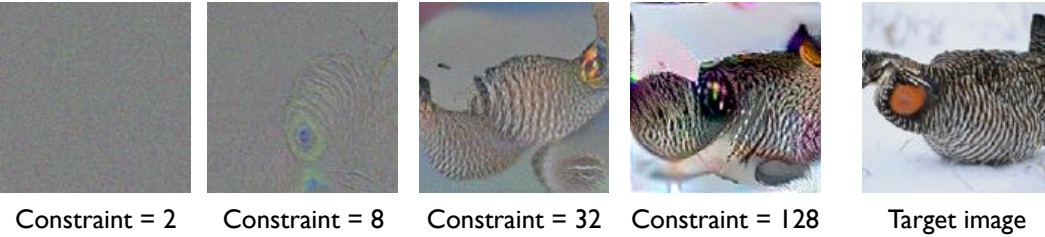

Figure 5: A visualization of the final solutions to the optimizing objective (4) with PGD when constraining the solution to lie in an $\ell_2$ ball around the source image for an adversarially robust neural network. We note that even the radius of the constraint set is small and we cannot match the representation very well, salient features of the target image still arise.

We also find that even when $\delta$ is highly constrained (i.e. when we are forced to stay very close to the source image and thus cannot match the representation of the target well), the solution to the inversion problem still displays some salient features of the target image (c.f. Figure 5). Both of these observations suggest that the representations of robust networks function much more like we would expect high-level feature representations to behave.

**Inversion of out-of-distribution inputs.** We find that the inversion properties uncovered above hold even for out-of-distribution inputs, demonstrating that robust representations capture *general* features as opposed to features only relevant for the specific classification task. In particular, we repeat the inversion experiment (simple minimization of distance in representation space) using images from classes not present in the original dataset used during training (Figure 6 right) and structured random patterns (Figure 14 in Appendix B.1): the reconstructed images consistently resemble the targets.

**Interpolation between arbitrary inputs.** Note that this ability to consistently invert representations into corresponding inputs also translates into the ability to *semantically interpolate* between any two inputs. For any two inputs $x_1$ and $x_2$, one can (linearly) interpolate between $R(x_1)$ and $R(x_2)$ in representation space, then use the inversion procedure to get images corresponding to the interpolate representations. The resulting inputs interpolate between the two endpoints in a percep-

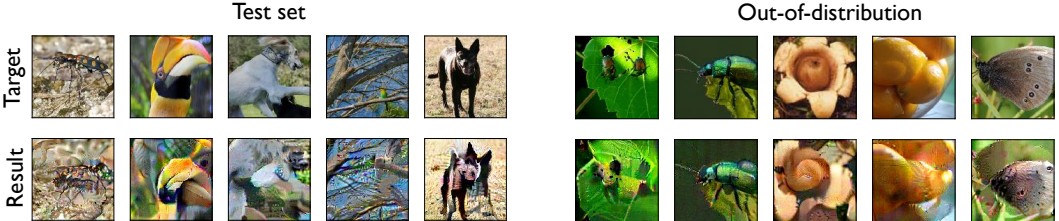

Figure 6: Robust representations yield semantically meaningful embeddings. *Target*: random images from the test set (col. 1-5) and from outside of the training distribution (6-10); *Result*: images obtained from optimizing inputs (using Gaussian noise as the source image) to minimize $\ell_2$-distance to the representations of the corresponding image in the top row. (More examples appear in Appendix B.1.)

tually plausible manner without any of the "ghosting" artifacts present in input-space interpolation. We show examples of this inversion as well as experimental details in Appendix A.5.

## 4.2 DIRECT FEATURE VISUALIZATION

A common technique for visualizing and understanding the representation function $R(\cdot)$ of a given network is *optimization-based feature visualization* (Olah et al., 2017), a process in which we maximize a specific feature (component) in the representation with respect to the input, in order to obtain insight into the role of the feature in classification. Concretely, given some $i \in [k]$ denoting a component of the representation vector, we use gradient descent to find an input $x'$ that maximally activates it, i.e., we solve:

$$x' = x_0 + \arg\max_{\delta} R(x_0 + \delta)_i \tag{5}$$

for various starting points $x_0$ which might be random images from $\mathcal{D}$ or even random noise.

**Visualization "fails" for standard networks.** For standard networks, optimizing the objective (5) often yields unsatisfying results. While we *can* easily find images for which the $i^{th}$ component of $R(\cdot)$ is large (and thus the optimization problem is tractable), these images tends to look meaningless to humans, often resembling the starting point of the optimization. Even when these images are non-trivial, they tend to contain abstract, hard-to-discern patterns (c.f. Figure 7 (bottom)). As we discuss later in this section, regularization/post-processing of visualizations does improve this state of affairs, though not very significantly and potentially at the cost of suppressing useful features present in the representation post-hoc.

**Robust representations allow for direct visualization of human-recognizable features.** For robust representations, however, we find that easily recognizable high-level features emerge from optimizing objective (5) directly, *without any regularization or post-processing*. We present the results of this maximization in Figure 7 (top): coordinates consistently represent the same concepts across different choice of starting input $x_0$ (both in and out of distribution). Furthermore, these concepts are not merely an artifact of our visualization process, as they consistently appear in the test-set inputs that most strongly activate their corresponding coordinates (Figure 8).

**The limitations of regularization for visualization in standard networks.** Given that directly optimizing objective (5) does not produce human-meaningful images, prior work on visualization usually tries to regularize objective (5) through a variety of methods. These methods include applying random transformations during the optimization process (Mordvintsev et al., 2015; Olah et al., 2017), restricting the space of possible solutions (Nguyen et al., 2015; 2016; 2017), or post-processing the input or gradients (Oygard, 2015; Tyka, 2016). While regularization does in general produce better results qualitatively, it comes with a few notable disadvantages that are well-recognized in the domain of feature visualization. First, when one introduces prior information about what makes images visually appealing into the optimization process, it becomes difficult to disentangle the effects of the actual model from the effect of the prior information introduced

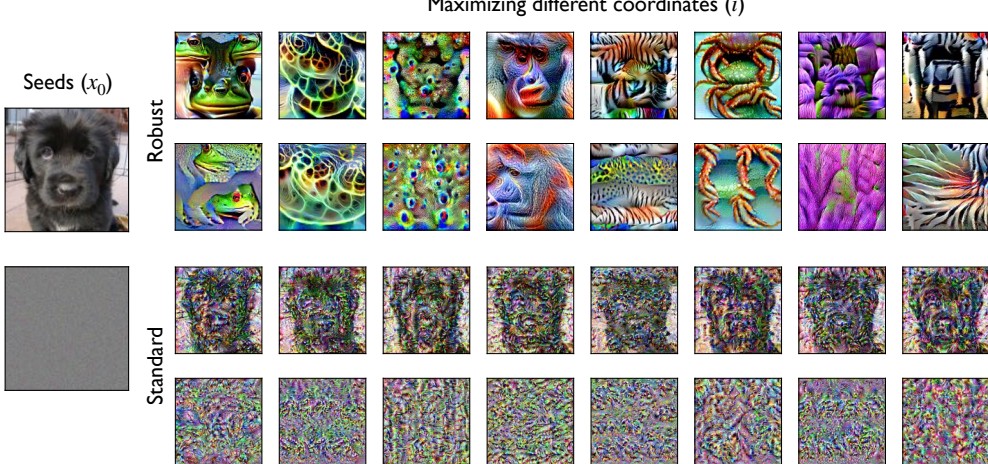

Figure 7: Correspondence between image-level patterns and activations learned by standard and robust models on the Restricted ImageNet dataset. Starting from randomly chosen seed inputs (noise/images), we use PGD to find inputs that (locally) maximally activate a given component of the representation vector (cf. Appendix A.6.1 for details). In the left column we have the seed inputs $x_0$ (selected *randomly*), and in subsequent columns we visualize the result of the optimization (5), i.e., $x'$, for different activations, with each row starting from the same (far left) input $x_0$ for (*top*): a robust (adversarially trained) and (*bottom*): a standard model. Additional visualizations in Appendix B.3, and similar results for ImageNet in B.3.2.

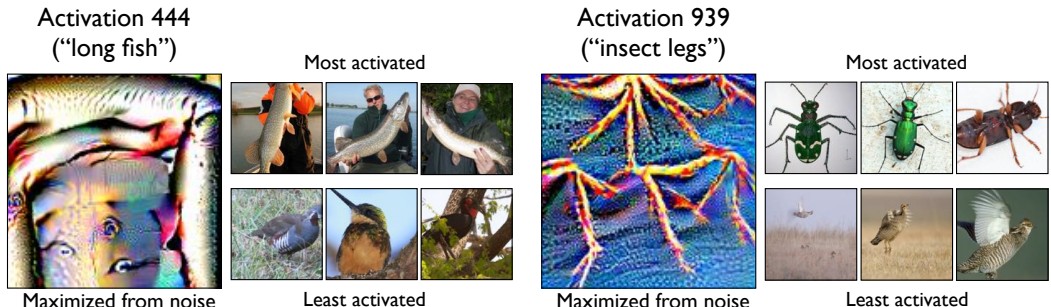

Figure 8: Maximizing inputs $x'$ (found by solving (5) with $x_0$ being a gray image) and most or least activating images (from the test set) for two *random* activations of a robust model trained on the Restricted ImageNet dataset. For each activation, we plot the three images from the validation set that had the highest or lowest activation value sorted by the magnitude of the selected activation.

through regularization[2]. Furthermore, while adding regularization does improve the visual quality of the visualizations, the components of the representation still cannot be shown to correspond to any recognizable high-level feature. Indeed, Olah et al. (2017) note that in the representation layer of a standard GoogLeNet, "Neurons do not seem to correspond to particularly meaningful semantic ideas"—the corresponding feature visualizations are reproduced in Figure 9. We also provide examples of representation-layer visualizations for VGG16 (which we found qualitatively best among modern architectures) regularized with jittering and random rotations in Figure 10. While these visualizations certainly look better qualitatively than their unregularized counterparts in Figure 7 (bottom), there remains a significantly large gap in quality and discernability between these regularized visualizations and those of the robust network in Figure 7 (top).

---

[2]In fact, model explanations that enforce priors for purposes of visual appeal have been often found to have little to do with the data or the model itself (Adebayo et al., 2018).

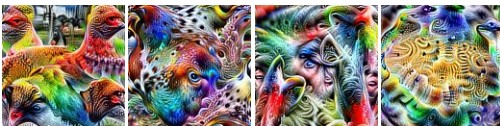

Figure 9: Figure reproduced from (Olah et al., 2017)—a visualization of a few components of the representation layer of GoogLeNet. While regularization (as well as Fourier parameterization and colorspace decorrelation) yields visually appealing results, the visualization does not reveal consistent semantic concepts.

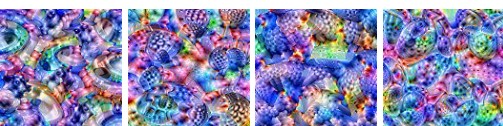

Figure 10: A visualization of the first four components of the representation layer of VGG16 when regularization via random jittering and rotation is applied. Figure produced using the Lucid[a] visualization library.

------

[a]https://github.com/tensorflow/lucid/

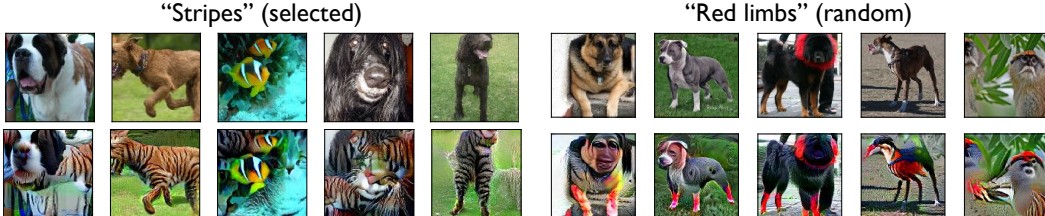

Figure 11: Visualization of the results from maximizing a chosen (left) and a *random* (right) representation coordinate starting from *random* images for the Restricted ImageNet dataset. In each figure, the top row has the initial images, and the bottom row has a feature added. Additional examples in Appendix B.4.

#### 4.2.1 NATURAL CONSEQUENCE: FEATURE MANIPULATION

The ability to directly visualize high-level, recognizable features reveals another application of robust representations, which we refer to as *feature manipulation*. Consider the visualization objective (5) shown in the previous section. Starting from some original image, optimizing this objective results in the corresponding feature being introduced in a continuous manner. It is hence possible to stop this process relatively early to ensure that the content of the original image is preserved. As a heuristic, we stop the optimization process as soon as the desired feature attains a larger value than all the other coordinates of the representation. We visualize the result of this process for a variety of input images in Figure 11, where *"stripes"* or *"red limbs"* are introduced seamlessly into images without any processing or regularization [3].

## 5 RELATED WORK

**Adversarial Robustness** Our work studies the feature representations of *adversarially robust networks*. As discussed in Section 3, these are networks trained with the robust optimization framework (Wald, 1945; Goodfellow et al., 2015; Madry et al., 2018) and were originally proposed in the context of defending against adversarial perturbations (Biggio et al., 2013; Szegedy et al., 2014). Adversarial robustness has been studied extensively in the context of machine learning security (see e.g., Carlini & Wagner (2017); Athalye et al. (2018b;a); Papernot et al. (2017)), and as an independent phenomenon (see e.g., Gilmer et al. (2018); Schmidt et al. (2018); Jacobsen et al. (2019); Ilyas et al. (2019); Tsipras et al. (2019); Su et al. (2018). Recent work also uses robust models for input manipulation: Tsipras et al. (2019) observe that large adversarial perturbation constructed for robust networks actually resemble instances of the target class, and Anon. (2019)[4] demonstrates that robust classifiers can be used for a wide array of image synthesis tasks. While our work also manipulates inputs with robust classifiers, we focus on understanding properties of robust representations (via inversion and component visualization), rather than perform any downstream tasks.

------

[3]We repeat this process with many additional random images and random features in Appendix B.4.

[4]Anonymized for the rebuttal stage.

**Inverting representations.** Previous methods for inverting learned representations typically either solve an optimization problem similar to (1) while imposing a "natural image" prior on the input Mahendran & Vedaldi (2015); Yosinski et al. (2015); Ulyanov et al. (2017) or train a separate model to perform the inversion Kingma & Welling (2015); Dosovitskiy & Brox (2016b;a). Note that since these methods introduce priors or additional components into the inversion process, their results are not fully faithful to the model. In an orthogonal direction, it is possible to construct models that are analytically invertible by construction Dinh et al. (2014; 2017); Jacobsen et al. (2018); Behrmann et al. (2018). However, the representations learned by these models do not seem to be perceptually meaningful (for instance, interpolating between points in the representation space does not lead to perceptual input space interpolations Jacobsen et al. (2018)). Another notable distinction between the inversions shown here and invertible networks is that the latter are an exactly invertible map from $\mathbb{R}^d \to \mathbb{R}^d$, while the former shows that we can approximately recover the original input in $\mathbb{R}^d$ from a representation in $\mathbb{R}^k$ for $k \ll d$.

**Feature visualization.** Typical methods for visualizing features or classes learned by deep networks follow an optimization-based approach, optimizing objectives similar to objective (5). Since this optimization does not lead to meaningful visualizations directly, these methods incorporate domain-specific input priors (either hand-crafted Nguyen et al. (2015) or learned Nguyen et al. (2016; 2017)) and regularizers Simonyan et al. (2013); Mordvintsev et al. (2015); Oygard (2015); Yosinski et al. (2015); Tyka (2016); Olah et al. (2017) to produce human-discernible visualizations. The key difference of our work is that we avoid the use of such priors or regularizers altogether, hence producing visualizations that are fully based on the model itself without introducing any additional bias.

**Semantic feature manipulation.** The latent space of generative adversarial networks (GANs) Goodfellow et al. (2014) tends to allow for "semantic feature arithmetic" Radford et al. (2016); Larsen et al. (2016) (similar to that in word2vec embeddings Mikolov et al. (2013)) where one can manipulate salient input features using latent space manipulations. In a similar vein, one can utilize an image-to-image translation framework to perform such manipulation (e.g. transforming horses to zebras), although this requires a task-specific dataset and model Zhu et al. (2017). Somewhat orthogonally, it is possible to utilize the deep representations of *standard* models to perform semantic feature manipulations; however such methods tend to either only perform well on datasets where the inputs are center-aligned Upchurch et al. (2017), or are restricted to a small set of manipulations Gatys et al. (2016).

## 6 CONCLUSION

We show that the learned representations of robustly trained models align much more closely with our idealized view of neural network embeddings as extractors of human-meaningful, high-level features. After highlighting certain shortcomings of standard deep networks and their representations, we demonstrate that robust optimization can actually be viewed as inducing a *human prior* over the features that models are able to learn. In this way, one can view the *robust representations* that result from this prior as feature extractors that are more aligned with human perception.

In support of this view, we demonstrate that robust representations overcome the challenges identified for standard representations: they are approximately invertible, and moving towards an image in representation space seems to entail recovering salient features of that image in pixel space. Furthermore, we show that robust representations can be directly visualized with first-order methods without the need for post-processing or regularization, and also yield much more human-understandable features than standard models (even when they are visualized with regularization). These two properties (inversion and direct feature visualization), in addition to serving as illustrations of the benefits of robust representations, also enable direct modes of input manipulation (interpolation and feature manipulation, respectively).

Overall, our findings highlight robust optimization as a framework to enforce feature priors on learned models. We believe that further exploring this paradigm will lead to models that are significantly more human-aligned while enabling a wide range of new modes of interactions.

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

# A    EXPERIMENTAL SETUP

## A.1    DATASETS

In the main text, we perform all our experimental analysis on the Restricted ImageNet dataset Tsipras et al. (2019) which is obtained by grouping together semantically similar classes from ImageNet into 9 super-classes shown in Table 1. Attaining robust models for the complete ImageNet dataset is known to be challenging, both due to the hardness of the learning problem itself, as well as the computational complexity.

For the sake of completeness, we also replicate our experiments feature visualization and representation inversion on the complete ImageNet dataset Russakovsky et al. (2015) in Appendices B.3.2 and B.1.4—in particular, cf. Figures 22 and 16.

Table 1: Classes used in the Restricted ImageNet model. The class ranges are inclusive.

| Class | Corresponding ImageNet Classes |
|---|---|
| "Dog" | 151 to 268 |
| "Cat" | 281 to 285 |
| "Frog" | 30 to 32 |
| "Turtle" | 33 to 37 |
| "Bird" | 80 to 100 |
| "Primate" | 365 to 382 |
| "Fish" | 389 to 397 |
| "Crab" | 118 to 121 |
| "Insect" | 300 to 319 |

## A.2    MODELS

We use the ResNet-50 architecture He et al. (2016) for our adversarially trained classifiers on all datasets. Unless otherwise specified, we use standard ResNet-50 classifiers trained using empirical risk minimization as a baseline in our experiments. Additionally, it has been noted in prior work that among standard classifiers, VGG networks Simonyan & Zisserman (2015) tend to have better-behaved representations and feature visualizations Mordvintsev et al. (2018). Thus, we also compare against standard VGG16 networks in the subsequent appendices. All models are trained with data augmentation, momentum 0.9 and weight decay $5e^{-4}$. Other hyperparameters are provided in Tables 2 and 3.

The exact procedure used to train robust models along with the corresponding hyperparameters are described in Section A.3. For standard (not adversarially trained) classifiers on the complete 1k-class ImageNet dataset, we use pre-trained models provided in the PyTorch repository[5].

Table 2: Standard hyperparameters for the models trained in the main paper.

| Dataset | Model | Arch. | Epochs | LR | Batch Size | LR Schedule |
|---|---|---|---|---|---|---|
| Restricted ImageNet | standard | ResNet-50 | 110 | 0.1 | 256 | Drop by 10 at epochs $\in [30, 60]$ |
| Restricted ImageNet | robust | ResNet-50 | 110 | 0.1 | 256 | Drop by 10 at epochs $\in [30, 60]$ |
| ImageNet | robust | ResNet-50 | 110 | 0.1 | 256 | Drop by 10 at epochs $\in [100]$ |

Test performance of all the classifiers can be found in Section A.4. Specific parameters used to study the properties of learned representations are described in Section A.6.

---

[5]https://pytorch.org/docs/stable/torchvision/models.html

### A.3 ADVERSARIAL TRAINING

To obtain robust classifiers, we employ the adversarial training methodology proposed in Madry et al. (2018). Specifically, we train against a projected gradient descent (PGD) adversary with a normalized step size, starting from a random initial perturbation of the training data. We consider adversarial perturbations in $\ell_2$-norm. Unless otherwise specified, we use the values of $\epsilon$ provided in Table 3 to train/evaluate our models (the images themselves lie in the range $[0, 1]$).

Table 3: Hyperparameters used for adversarial training.

| Dataset | $\epsilon$ | # steps | Step size |
|---|---|---|---|
| Restricted ImageNet | 3.0 | 7 | 0.5 |
| ImageNet | 3.0 | 7 | 0.5 |

### A.4 MODEL PERFORMANCE

Standard test performance for the models used in the paper are presented in Table 4 for the Restricted ImageNet dataset and in Table 5 for the complete ImageNet dataset.

Additionally, we report adversarial accuracy of both standard and robust models. Here, adversarial accuracies are computed against a PGD adversary with 20 steps and step size of $0.375$. (We also evaluated against a stronger adversary using more steps (100) of PGD, however this had a marginal effect on the adversarial accuracy of the models.)

Table 4: Test accuracy for standard and robust models on the Restricted ImageNet dataset.

| Model | Standard | Adversarial (eps=3.0) |
|---|---|---|
| Standard VGG16 | 98.22% | 2.17% |
| Standard ResNet-50 | 98.01% | 4.74% |
| Robust ResNet-50 | 92.39% | 81.91% |

Table 5: Top-1 accuracy for standard and robust models on the ImageNet dataset.

| Model | Standard | Adversarial (eps=3.0) |
|---|---|---|
| Standard VGG16 | 73.36% | 0.35% |
| Standard ResNet-50 | 76.13% | 0.13% |
| Robust ResNet-50 | 57.90% | 35.16% |

### A.5 IMAGE INTERPOLATIONS

A natural consequence of the "natural invertibility" property of robust representations is the ability to synthesize natural interpolations between any two inputs $x_1, x_2 \in \mathbb{R}^n$. In particular, given two images $x_1$ and $x_2$, we define the $\lambda$-*interpolate* between them as

$$x_\lambda = \min_x \| (\lambda \cdot R(x_1) + (1 - \lambda) \cdot R(x_2)) - R(x) \|_2. \tag{6}$$

where, for a given $\lambda$, we find $x_\lambda$ by solving (6) with projected gradient descent. Intuitively, this corresponds to linearly interpolating between the points in representation space and then finding a point in image space that has a similar representation. To construct a length-$(T + 1)$ interpolation, we choose $\lambda = \{0, \frac{1}{T}, \frac{2}{T}, \dots 1\}$. The resulting interpolations, shown in Figure 12, demonstrate that the $\lambda$-interpolates of robust representations correspond to a meaningful feature interpolation between images. (For standard models constructing meaningful interpolations is impossible due to the brittleness identified in Section 2—see Appendix B.1.3 for details.)

**Top: Image-space interpolation**

dog → dog      **Bottom: Representation-space interpolation**      random endpoints

Figure 12: Image interpolation using robust representations compared to their image-space counterparts. The former appear perceptually plausible while the latter exhibit ghosting artifacts. For pairs of images from the Restricted ImageNet test set, we solve (6) for $\lambda$ varying between zero and one, i.e., we match linear interpolates in representation space. Additional interpolations appear in Appendix B.2.1 Figure 17. We demonstrate the ineffectiveness of interpolation with standard representations in Appendix B.2.2 Figure 18.

**Relation to other interpolation methods.** We emphasize that linearly interpolating in robust representation space works for *any* two images. This generality is in contrast to interpolations induced by GANs (e.g. (Radford et al., 2016; Brock et al., 2019)), which can only interpolate between images generated by the generator. (Reconstructions of out-of-range images tend to be decipherable but rather different from the originals Bau et al. (2019).) It is worth noting that even for models with analytically invertible representations, interpolating in representation space does not yield semantic interpolations Jacobsen et al. (2018).

## A.6   PARAMETERS USED IN STUDIES OF ROBUST/STANDARD REPRESENTATIONS

### A.6.1   FINDING REPRESENTATION-FEATURE CORRESPONDENCE

| Dataset | $\epsilon$ | # steps | Step size |
|---|---|---|---|
| Restricted ImageNet/ImageNet | 1000 | 200 | 1 |

### A.6.2   INVERTING REPRESENTATIONS AND INTERPOLATIONS

| Dataset | $\epsilon$ | # steps | Step size |
|---|---|---|---|
| Restricted ImageNet/ImageNet | 1000 | 10000 | 1 |

# B OMITTED FIGURES

## B.1 INVERTING REPRESENTATIONS

### B.1.1 RECOVERING TEST SET IMAGES USING ROBUST REPRESENTATIONS

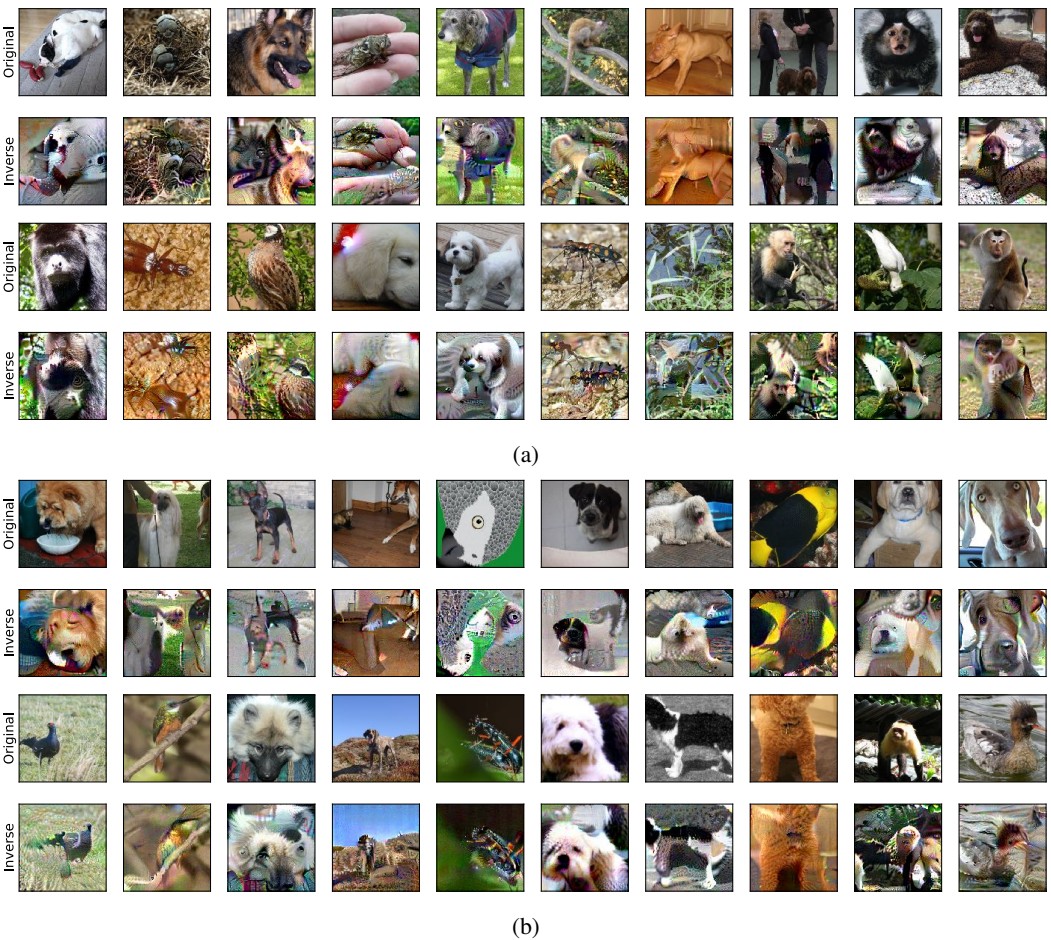

Figure 13: Robust representations yield semantically meaningful inverses: *Original*: randomly chosen test set images from the Restricted ImageNet dataset; *Inverse*: images obtained by inverting the representation of the corresponding image in the top row by solving the optimization problem (1) starting from: (a) different test images and (b) Gaussian noise.

### B.1.2 RECOVERING OUT-OF-DISTRIBUTION INPUTS USING ROBUST REPRESENTATIONS

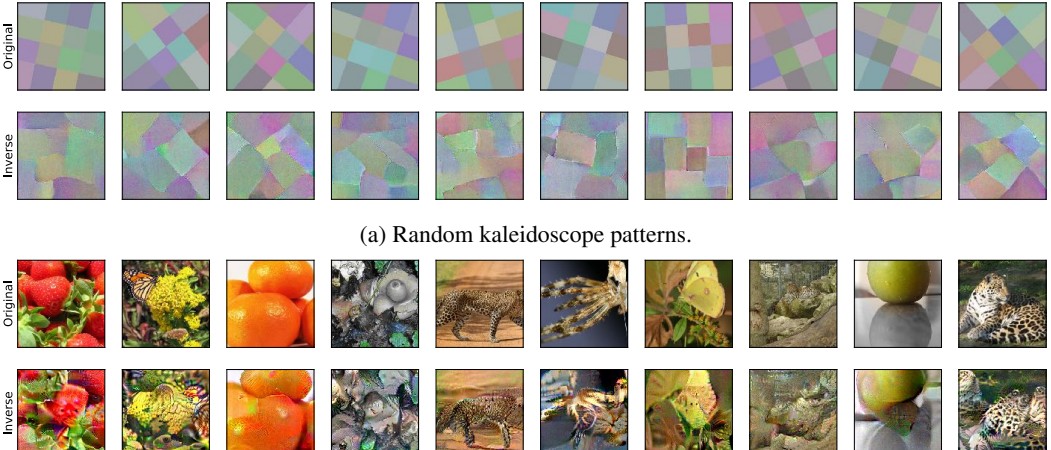

(a) Random kaleidoscope patterns.

(b) Samples from other ImageNet classes outside what the model is trained on.

Figure 14: Robust representations yield semantically meaningful inverses: (*Original*): randomly chosen out-of-distribution inputs; (*Inverse*): images obtained by inverting the representation of the corresponding image in the top row by solving the optimization problem (1) starting from Gaussian noise.

### B.1.3 INVERTING STANDARD REPRESENTATIONS

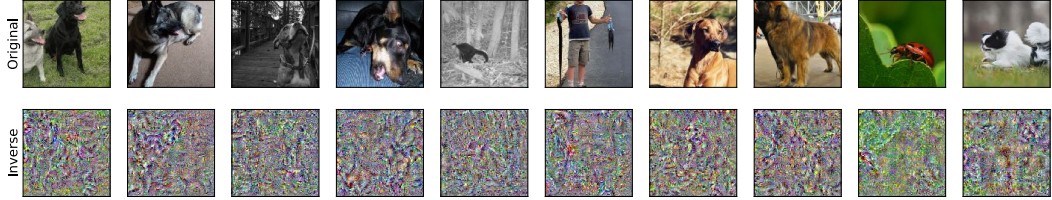

Figure 15: Standard representations *do not* yield semantically meaningful inverses: (*Original*): randomly chosen test set images from the Restricted ImageNet dataset; (*Inverse*): images obtained by inverting the representation of the corresponding image in the top row by solving the optimization problem (1) starting from Gaussian noise.

### B.1.4 REPRESENTATION INVERSION ON THE IMAGENET DATASET

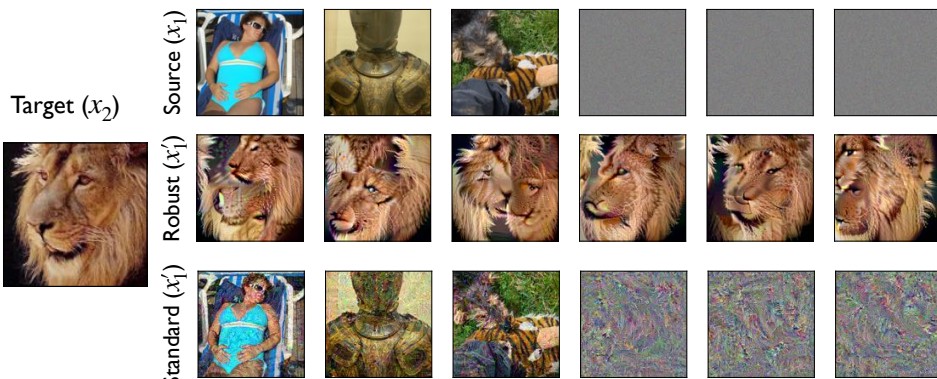

Figure 16: Visualization of inputs that are mapped to similar representations by models trained on the ImageNet dataset. *Target ($x_2$) & Source ($x_1$)*: random examples image from the test set; *Robust and Standard ($x_1'$)*: result of minimizing the objective (4) to match (in $\ell_2$-distance) the representation of the target image starting from the corresponding source image for (*top*): a robust (adversarially trained) and (*bottom*): a standard model respectively. For the robust model, we observe that the resulting images are perceptually similar to the target image in terms of high-level features, while for the standard model they often look more similar to the source image which is the seed for the optimization process.

## B.2 IMAGE INTERPOLATIONS

### B.2.1 ADDITIONAL INTERPOLATIONS FOR ROBUST MODELS

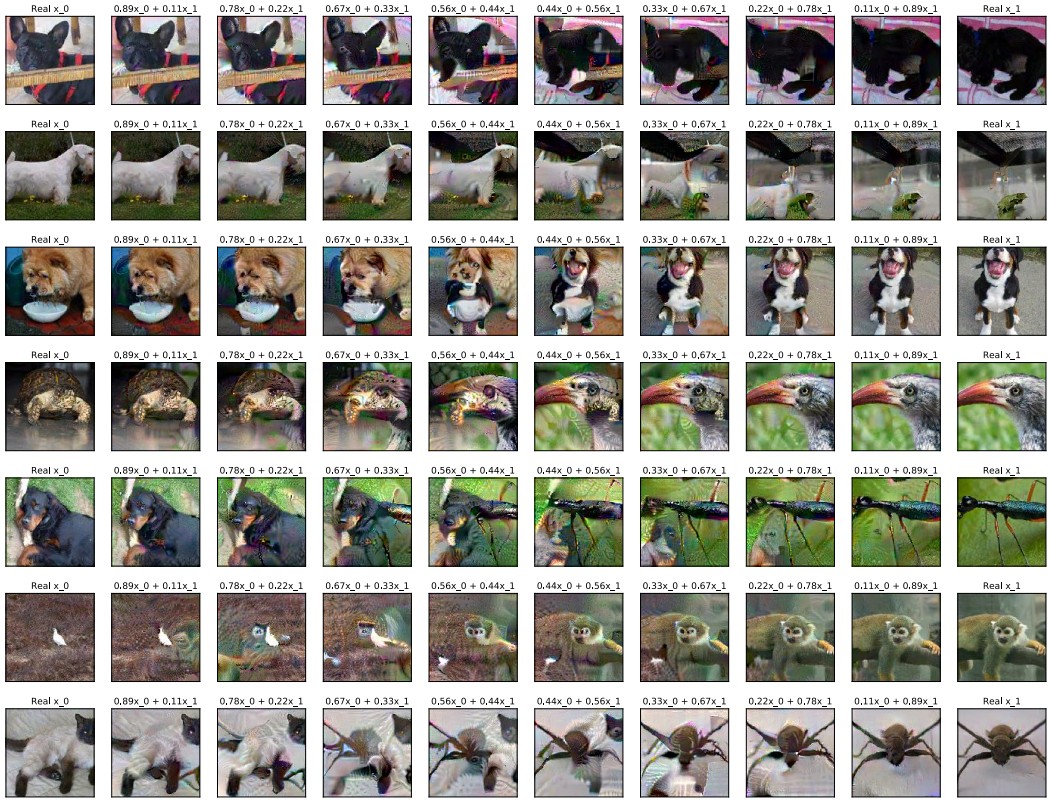

Figure 17: Additional image interpolation using robust representations. To find the interpolation in input space, we construct images that map to linear interpolations of the endpoints in robust representation space. Concretely, for randomly selected pairs from the Restricted ImageNet test set, we use (1) to find images that match to the linear interpolates in representation space (6).

### B.2.2 INTERPOLATIONS FOR STANDARD MODELS

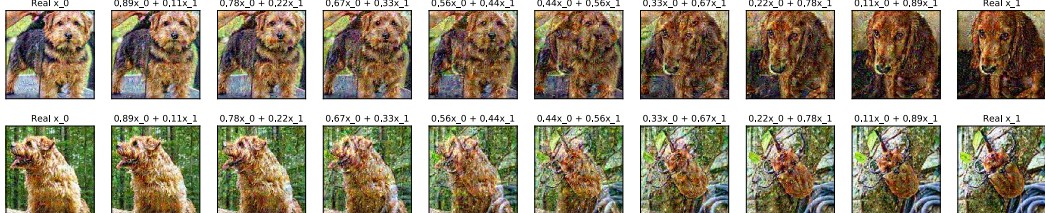

Figure 18: Image interpolation using standard representations. To find the interpolation in input space, we construct images that map to linear interpolations of the endpoints in standard representation space. Concretely, for randomly selected pairs from the Restricted ImageNet test set, we use (1) to find images that match to the linear interpolates in representation space (6). Image space interpolations from the standard model appear to be significantly less meaningful than their robust counterparts. They are visibly similar to linear interpolation directly in the input space, which is in fact used to seed the optimization process.

## B.3   DIRECT FEATURE VISUALIZATIONS FOR STANDARD AND ROBUST MODELS

### B.3.1   ADDITIONAL FEATURE VISUALIZATIONS FOR THE RESTRICTED IMAGENET DATASET

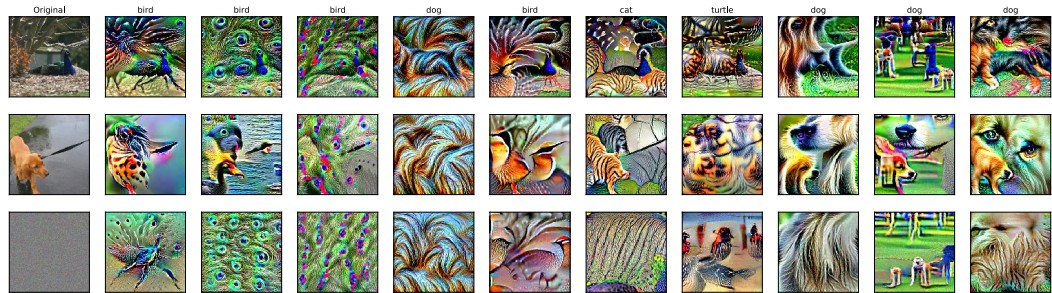

Figure 19: Correspondence between image-level features and representations learned by a robust model on the Restricted ImageNet dataset. Starting from randomly chosen seed inputs (noise/images), we use a constrained optimization process to identify input features that maximally activate a given component of the representation vector (cf. Appendix A.6.1 for details). Specifically, (*left column*): inputs to the optimization process, and (*subsequent columns*): features that activate randomly chosen representation components, along with the predicted class of the feature.

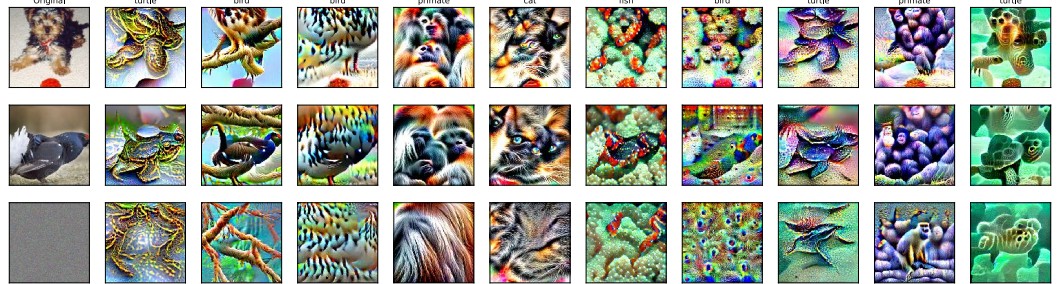

Figure 20: Correspondence between image-level features and representations learned by a robust model on the Restricted ImageNet dataset. Starting from randomly chosen seed inputs (noise/images), we use a constrained optimization process to identify input features that maximally activate a given component of the representation vector (cf. Appendix A.6.1 for details). Specifically, (*left column*): inputs to the optimization process, and (*subsequent columns*): features that activate select representation components, along with the predicted class of the feature.

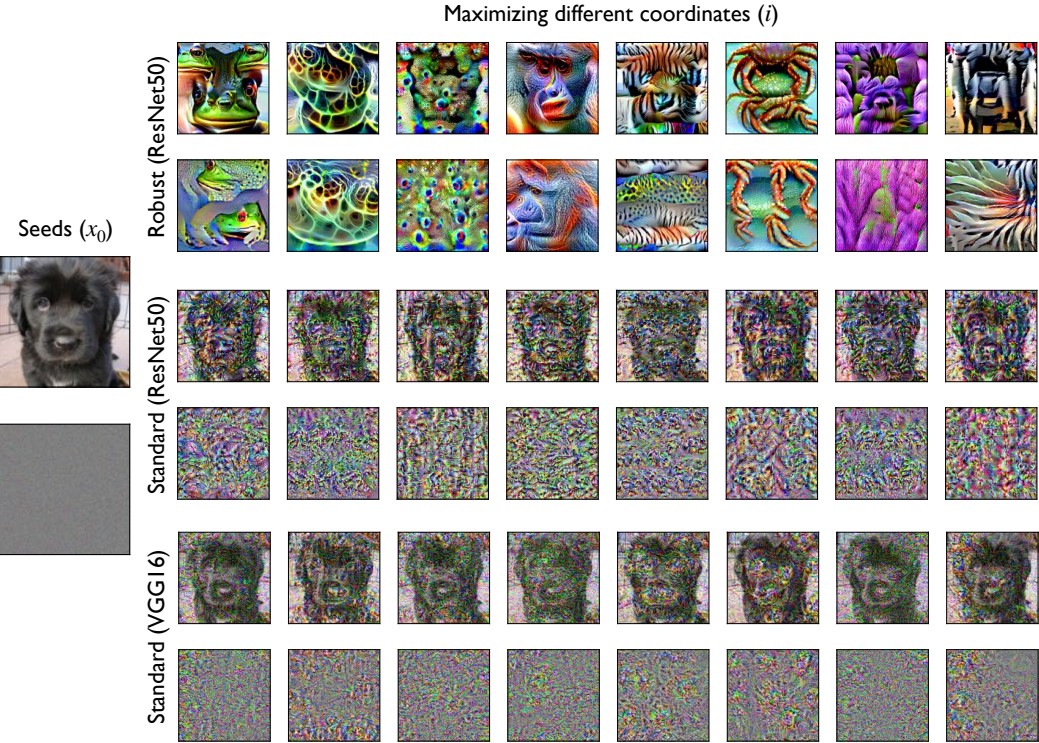

Figure 21: Correspondence between image-level patterns and activations learned by standard and robust models on the Restricted ImageNet dataset. Starting from randomly chosen seed inputs (noise/images), we use PGD to find inputs that (locally) maximally activate a given component of the representation vector (cf. Appendix A.6.1 for details). In the left column we have the original inputs (selected *randomly*), and in subsequent columns we visualize the result of the optimization (5) for different activations, with each row starting from the same (far left) input for (*top*): a robust (adversarially trained) ResNet-50 model, (*middle*): a standard ResNet-50 model and (*bottom*): a standard VGG16 model.

### B.3.2 FEATURE VISUALIZATIONS FOR THE IMAGENET DATASET

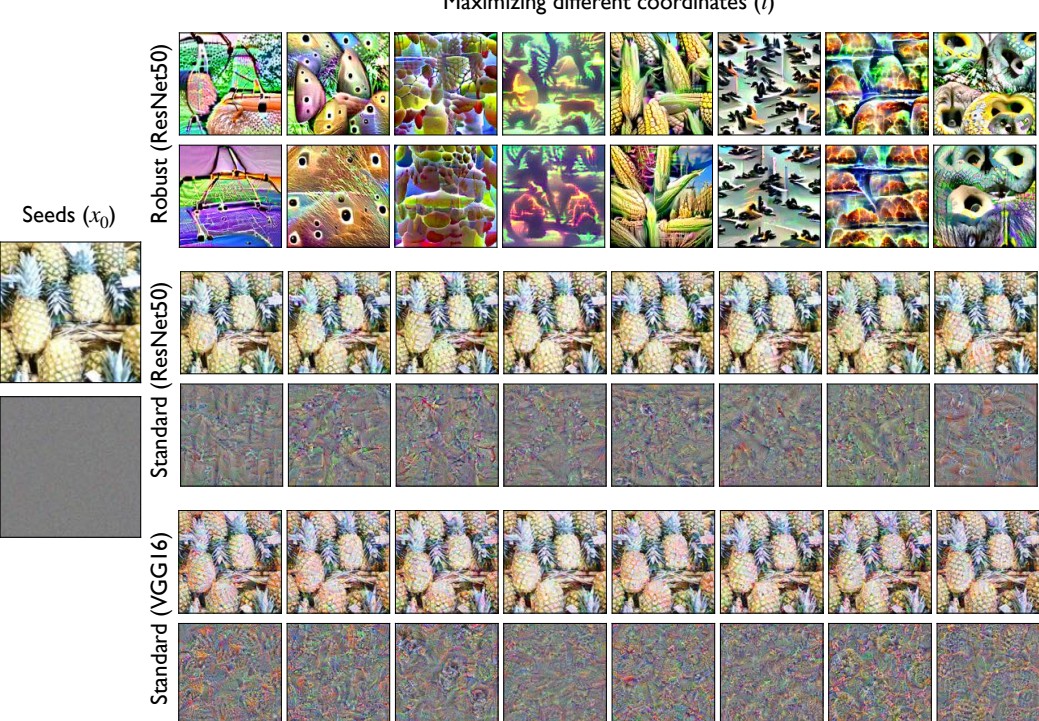

Figure 22: Correspondence between image-level patterns and activations learned by standard and robust models on the complete ImageNet dataset. Starting from randomly chosen seed inputs (noise/images), we use PGD to find inputs that (locally) maximally activate a given component of the representation vector (cf. Appendix A.6.1 for details). In the left column we have the original inputs (selected *randomly*), and in subsequent columns we visualize the result of the optimization (5) for different activations, with each row starting from the same (far left) input for (*top*): a robust (adversarially trained) ResNet-50 model, (*middle*): a standard ResNet-50 model and (*bottom*): a standard VGG16 model.

## B.4 Additional examples of feature manipulation

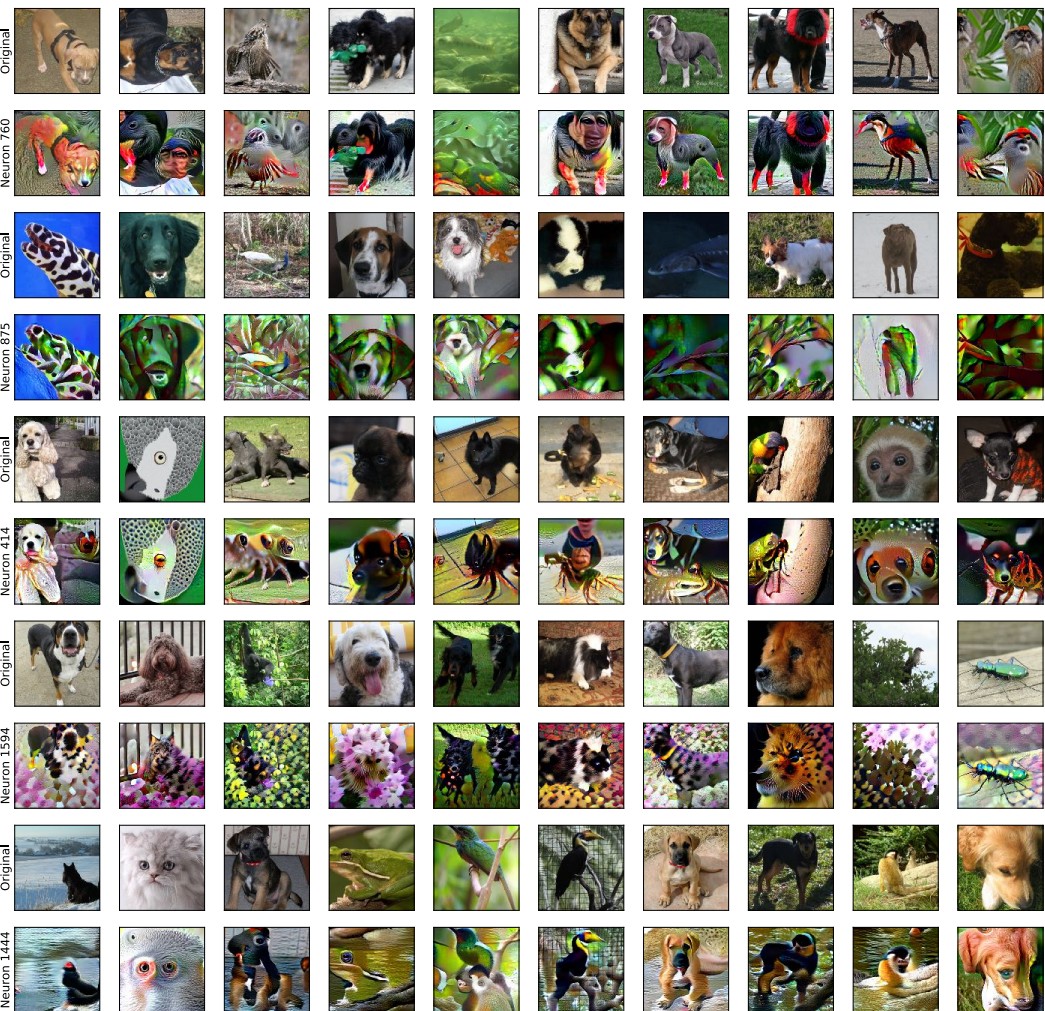

Figure 23: Visualization of the results adding various neurons, labelled on the left, to randomly chosen test images. The rows alternate between the original test images, and those same images with an additional feature arising from maximizing the corresponding neuron.

