# OpenReview forum: "Adversarial Robustness as a Prior for Learned Representations"
_ICLR.cc/2020/Conference — Reject_

### Official Review · AnonReviewer1 · 2019-10-20
**Official Blind Review #1**

**Rating:** 6

**Review:**

Summary:
The paper shows that the learnt representations of robustly trained models align more closely with features that the human perceive as meaningful. They propose that robust optimization can be viewed as inducing a human prior over learnt features. Extensive experiments demonstrate that robust representations are approximately invertible, can be visualized yielding more human-interpretable features, and enable direct modes of input manipulations.

The paper indicate adversarial robustness as a promising avenue for improving learned representations in from several aspects. It is well written and contains extension experimental results. I'd suggest accepting the paper.

Questions and Comments:
- Is there a particular reason that $L_{2}$ norm is used throughout the paper? How is the performance if using other ones?
- Compared to the other methods introducing priors or additional components into the inversion process, how is the quantitative inversion quality and computational complexity of the proposed method?
- The paper claims that the representations are more perceptually meaningful than the others, which may need to be evaluated with broader human subjective.
- I think it should be argmin in Equation (1).
- Also Equation (4) is exactly the same as (1).
- Some symbols seem to be used somewhat interchangeably. E.g., x' in Equation (1) represents x+\delta, while x' in Equation (5) is \delta itself.

**Experience Assessment:**

I have read many papers in this area.

**Review Assessment: Checking Correctness Of Derivations And Theory:**

I carefully checked the derivations and theory.

**Review Assessment: Checking Correctness Of Experiments:**

I carefully checked the experiments.

**Review Assessment: Thoroughness In Paper Reading:**

I read the paper thoroughly.

---

> ### Author Response · Authors · 2019-11-11
> **Response #1**
>
> We thank the reviewer for their comments and suggestions.
>
> Why $\ell_2$ robustness:
> In preliminary experiments, we found that both L-2 and L-infinity robust models (L2 and L-inf are the two most commonly studied threat models in adversarial robustness) yielded the properties explained in this paper.  We thus decided to only study one of these settings for simplicity., We do agree that taking a detailed look at the effect of robustness metric (L2, L-inf, etc.) on representations is an interesting future direction.
>
> Comparing our inversion results to other methods:
> From a computational perspective, our method does not require training a separate network or access to a generative model. As such, it would only be comparable to Mahendran & Veraldi (2015) who, like us, use gradient descent to minimize the L2 representation distance (Eq. 4). In comparison to theirs, our method does not require regularizers or hyper-parameter tuning and produces significantly better results qualitatively. Specifically, their inverted images often appear blurry or lack salient features of the original image (we invite the reviewers to inspect the results https://arxiv.org/abs/1412.0035). We attribute both the lack of need for regularization as well as the better qualitative results to the favorable properties of robust representations discussed in our work.
> From a performance perspective, given that the inversion quality for robust models outperforms that of standard models so clearly  (c.f. Figure 3 middle vs bottom), we decided that quantitative experiments would not be necessary to support the presented thesis.
>
> With respect to perceptual meaningfulness:
> It would indeed be interesting to see large-scale human studies comparing representation properties in future work. However, we believe that the difference between standard and robust networks (which is our main focus) is so apparent that a human-study would be unnecessary (see, for example, Figure 3 robust vs standard, or Figure 7 robust vs standard).
>
> We have also fixed the notational issues brought to our attention, we thank the reviewer for pointing them out:
> - Fixed (1) and (4) to be argmin instead of min (they are indeed the same equation but reproduced for clarity)
> - Equation (5) was missing an “x_0 + “ so that x always represents images.

---

### Official Review · AnonReviewer3 · 2019-10-23
**Official Blind Review #3**

**Rating:** 3

**Review:**

###  Summary
1. The paper proposes robustness to small adversarial perturbations as a prior when learning representations.
2. It demonstrates that representations that satisfy such a prior have non-trivial properties -- they are easier to visualize, are invertible (i.e. optimizing an input that produces the desired activation leads to reasonable images), and allows for direct manipulation of input features (by changing a feature in the representation space and then optimizing the image to satisfy this new representation.)

### Non-blind review
THIS IS NOT A BLIND REVIEW
Reviewing this paper reminded me of a recent NeurIPS paper I read.

I went back to that NeurIPS paper (to better compare the similarities and differences) only to found out:

1- The NeurIPS19 paper cites an earlier arxiv version of this paper as an inspiration for its approach.
2- It is from the exact same authors.

This, unfortunately, means I know who the authors are (however, there is no conflict of interest).

More importantly, this paper is too similar to the NeurIPS paper and It's hard to review without taking into account the NeurIPS paper. In this review, I will treat the said NeurIPS19 paper as published work, and evaluate if this work adds more to the discourse. (I've refrained from naming the neurips paper so the anonymity is maintained for other reviewers; the authors, I presume, would immediately know which paper I'm referring to).

### Decisions with reasons

Even though I think the idea introduced in this paper is interesting, I would argue for rejecting this paper for the simple reason: It doesn't add much to the existing discourse.

Using the proposed framework (i.e. learning robust representations), it demonstrates two phenomena.

First, it shows that robust models allow feature inversion. Second, it shows that it's easily possible to directly visualize and manipulate features for such a model. (Both of these are achieved using the same idea: treating input to the model as parameterized, and optimizing for a target activation)

These are interesting observations and show that robust models learn features that rely on salient parts of the input image. However, the NeurIPS19 paper shows this even more cleary.

As a result, I'm not convinced that demonstrating the same phenomena with different examples is sufficient for this to be a standalone paper. (Perhaps the two papers could have been one single paper).

### Questions

What is the rationale behind dividing examples showing robust models rely on salient parts of input into two papers? Is there a semantic meaning to the grouping i.e. showing feature inversion, feature manipulation, and visualization in one paper and Generation, inpainting, translation, etc in another?

If I understand correctly, all of these examples exist because the robustly learned representation relies on the salient parts of the input and not on the non-robust features. If that is the case, it makes more sense to show all of these examples in a single paper.

### Update after Author's response
Since the authors have added and discussed the pertinent NeurIPS paper in this submission, I'm updating my score.
I still think that the two papers are more similar than they might seem (See Re: Response 3 for more details).


### Update 2

I pointed out the similarities between the three contributions in this paper and the NeurIPS paper in "Re: Response #3" below. The authors replied to my concerns. I'm summarizing the author's position to my concerns followed by my response.

#### Author's Position
The authors agreed that features manipulation and feature visualization is similar, but pointed out that the chain of dependency is this paper -> NeurIPS paper and not the other way around. They mentioned that the NeurIPS paper cites this paper and acknowledges this. Moreover, they argued that even if we consider NeurIPS paper to be prior work, feature visualization is explored in much more detail in this paper.

#### Response
I think the direction of the chain of dependency is not that important since neither paper clearly builds on top of the other. The NeurIPS paper is published work now, and it makes sense to consider it prior work (Especially since it is from the same authors).

Moreover, during the NeurIPS review period, the authors did not cite this paper; they only added the citation in the camera-ready version. This means that during the NeurIPS review period, they did, in fact,  take credit for the ideas used in feature painting. (The authors mention that they somehow did not, and just stated the method and showed the pictorial result in the NeurIPS paper. However, I don't see how it is possible to present a method and a pictorial result without citing other work and not take credit for the method and result.)

I would agree with the authors that this paper does go into more detail for feature visualization. More specifically, this paper also looks at visualizing individual features in the representation (The NeurIPS feature painting restricts the visualization using a mask) and demonstrates that the same feature can be used to visualize similar semantic concepts (such as red limbs) on multiple images. This is definitely interesting, but still very related to the feature-painting result. It would have made more sense to include these feature visualization results in the NeurIPS paper instead of adding them in a separate paper.

#### Author's Position 2
They disagreed that feature inversion (this paper) is similar to image generation (NeurIPS paper). I did acknowledge in my initial response that feature inversion is slightly more general than image generation, however, the authors suggest that they are completely different."

#### Response
I think representation inversion is more similar to generation than it might seem. Representation for an in-distribution image would correspond to a class with high probability. Maximizing a class probability would indirectly optimize for a representation (Say R_0) that maximizes that class probability. Image generation, as presented in NeurIPS paper, can be seen as inverting R_0.

Moreover, the qualitative results for feature inversion, as presented in this paper, are not extra-ordinary. In the majority of the inverted images, I can not classify the inverted image correctly. That shows the model is still not paying attention to the correct aspects of the input to do classification. As a result, this paper certainly does not solve the feature inversion problem (Ideally, inverted features would highlight parts of the input necessary for making predictions and ignore other parts. Robust models, on the other hand, seem to be uniformly retaining all information of the image including the background and not highlighting the parts important for making predictions. As a result, many inverted images can not be classified by humans).

#### My current position
At the end of the day, both this and the NeurIPS paper are demonstration papers (They are empirically demonstrating an unintuitive phenomenon). Both papers are demonstrating that robust models learn features that correspond to salient parts of the input. Even though both papers are nice, either one is sufficient to demonstrate the phenomenon. For this to be a stand-alone paper, the authors would have to do more in my opinion. One option would be to explore and compare different forms of adversarial robustness as priors (The paper is called "Adversarial Robustness as a Prior for Learned Representations" and not "L2 Adversarial Robustness as a Prior for Learned Representations," after all). Another option would be to see if such representations are 'quantitatively' better in some settings (Such as for transfer learning).

In its current form, I feel that the two papers are too similar to recommend acceptance.

**Experience Assessment:**

I have read many papers in this area.

**Review Assessment: Checking Correctness Of Derivations And Theory:**

N/A

**Review Assessment: Checking Correctness Of Experiments:**

I carefully checked the experiments.

**Review Assessment: Thoroughness In Paper Reading:**

I read the paper thoroughly.

---

> ### Author Response · Authors · 2019-11-11
> **Response #3**
>
> # As per the AC's instructions, we have written this review assuming that everyone has access to the relevant paper (NeurIPS 2019).
>
> Both the NeurIPS 2019 paper and this submission manipulate inputs using a robust classifier. However, the papers are fundamentally different in almost every other facet:
>
> The NeurIPS 2019 paper is focused entirely on the task of image synthesis. The main contribution of that paper has nothing to do with representation learning, but rather demonstrates that tasks traditionally performed by generative models (or task-specific methods) can be accomplished with a classifier alone.
>
> In contrast, this work revolves around studying the features captured by the representations of robust classifiers, and showing that they are more aligned with human perception. Thus, our experiments are not on traditional computer vision tasks, but instead we use established methods for studying and understanding the representations learned by neural networks. Note that many of the tasks we consider have been the central studies of many representation learning papers:
>
> 1. Inversion: Previous work has established inversion of deep representations as in Section 4.1 of our work as a tool for understanding the features captured by the representation, e.g.,  Understanding Deep Image Representations by Inverting Them, (Mahendran & Vedaldi), and other references in our paper. Our experiments indicate that robust networks may be learning a much more human-aligned set of features, as inverting them actually approximately recovers the image without the need for any regularization or post-processing. (For standard networks, Mahendran & Vedaldi find that regularization/post-processing techniques are needed for even moderately decipherable results.)
>
> 2. Feature visualization & manipulation: Similarly, many prior works, e.g. Feature visualization (Olah et al) and others referenced in our paper, have established the feature visualization process as in Section 4.2 of our paper as a method for seeing what neurons are responsible for in classification. Our results show that in contrast to the negative result of Olah et al for standard networks (“Neurons [in the representation layer] do not seem to correspond to particularly meaningful semantic ideas”), neurons in the final layer of robust networks actually seem to learn clear, human-decipherable features.
>
> In summary, the goal of this submission is to study and understand the feature representations of robust network, and not to accomplish any sort of downstream task using the networks (the latter was precisely the goal of the NeurIPS 2019 paper).
>
> That said, since both papers do fall under the same broad umbrella of using a robust classifier to manipulate inputs, we should have referenced the NeurIPS 2019 paper from this one (failing to do so was a simple oversight on our part). We have updated the submission with a reference to the submission in the related work, and a shortened explanation of the difference between the two works.

---

> > ### Comment · AnonReviewer3 · 2019-11-11
> > **Re: Response #3**
> >
> > Thank you for your response.
> >
> > I do agree that the two papers focus on different things -- image synthesis is certainly not the same as representation inversion and feature manipulation; however, I don't agree they are fundamentally different.
> >
> > The results in the two papers are a corollary of a single observation -- features learned by robust model correspond to salient aspects of the inputs. For example:
> >
> > 1. Feature painting from the NeurIPS paper parallels features manipulation in this submission (with an added mask).
> >
> > 2. Feature visualization is also a corollary of feature painting -- in both cases, a feature is maximized with respect to the input image. The primary difference between the two is that in feature painting, the initial input image is a natural image (with a binary mask); whereas in feature visualization, the initial input image is a random image (or natural image without a mask).
> >
> > 3. Image Generation from the NeurIPS paper parallels representation inversion in this submission -- maximizing a target label indirectly corresponds to inverting representation that would maximize the target probability (Representations from in-domain images would already strongly correspond to a target label). I would acknowledge that feature inversion is slightly more general than image generation, and feature inversion of out-of-domain images is original to this paper. However, the qualitative results in B.1.2, Figure 14 are not very impressive -- I can not tell what any of the inverted images are without looking at the original images.
> >
> > For me, the primary appeal of both papers is that they are demonstrating an intuitive and interesting phenomenon on a range of examples. However, either of the two papers does a good job of demonstrating the phenomenon, and one doesn't add much to the discourse given the other.
> >
> > Nonetheless, since the NeurIPS paper has been added in the discussion of this paper, my concern now is only about the lack of novelty. I've improved my score by one increment to reflect that.

---

> > > ### Author Response · Authors · 2019-11-12
> > > **Response**
> > >
> > > Thank you for the clarification of your concerns.
> > >
> > > In our opinion, the primary issue here is the delineation between performing downstream applications, and understanding the learned representation. Crucially, performing these applications alone is not sufficient to make any statements about the quality of representations of robust networks (and vice-versa)—the former simply establishes, as the reviewer notes, that it is possible to change salient features in the input via the classifier. In order to study representations in this work, we turn to established representation learning tests like inversion and feature visualization, which are not implied by the results in the NeurIPS papers. For example, inversion is not implied by any results in the NeurIPS paper, as that paper only shows that the image is manipulatable using gradient descent in input space, and not that the features of the image can be recovered from the representation.
> > >
> > > Concretely, to respond to each of your numbered points:
> > >
> > > 1. Note that the NeurIPS paper cites the feature visualization observations from this submission as inspiration for feature painting.
> > >
> > > 2. See response (1) above. The correct chain of dependencies is thus (feature visualization [this work]) => (feature manipulation), (feature painting). Note that part of our goal in doing feature manipulation is again to probe properties of the representation, showing that that features are introduced *gradually* into the image. (Conversely, the NeurIPS paper is just trying to exploit both of these properties to accomplish the downstream task of interactive image editing).
> > >
> > > 3. We disagree that inversion parallels generation. For generation to work, all that is needed is that “maximizing the dog class in the network introduces dog features.” On the other hand, for inversion to work one needs that “the representation captures all the salient features of the given input in the representation.” In general, neither of these statements implies the other. (Also, in experiments for the NeurIPS paper we were unable to use inversion for generation, as we were unable to find representations corresponding to natural images).
> > >
> > > We hope that the above points have alleviated some of the reviewer’s concerns, and would be happy to elaborate on any of them further.

---

> > > > ### Comment · AnonReviewer3 · 2019-11-12
> > > > **Further clarification**
> > > >
> > > > It's not clear to me how the chain of dependency is this way. Was this paper submitted to NeurIPS concurrently to the Image Synthesis paper and didn't get in? Based on the dates they were first posted on arxiv, I presume that is the case.

---

> ### Comment · AnonReviewer3 · 2019-11-15
> **Review Updated**
>
> I've updated my review (See "Update 2").
>
> ** Response edited

---

> > ### Author Response · Authors · 2019-11-15
> > **Re: Review Updated**
> >
> > Regarding inversion vs generation: We stress that these two tasks are in fact entirely different. To elaborate on this, we illustrate that neither implies the other.
> >
> > -> Inversion does not imply generation: The reviewer claims that we can generate images by inverting some representation R_0. However, without explicitly saying how to find R_0, this is an entirely vacuous claim, as one could just say that any computer vision task is "just inverting some representation R_0" (after all, one could just solve the task, find the corresponding representation, and invert it). In our experiments, we tried finding R_0 by learning distributions over representations, perturbing representations of natural images, finding the representations that maximize class scores within a ball, and various other methods---we are actually unable to find *any* synthetic representation that can be inverted successfully.
> >
> > -> Generation does not imply inversion: This direction, which is __more important (as the current submission introduces inversion, not generation)__, is entirely clear: just because class maximization introduces salient features does *not* mean that the features captured by the representation are sufficient to approximately invert an image.
> >
> > As for the confusion around the prior work, we have stated numerous times that the lack of citation for the previous work in our submission to ICLR was an oversight and that the concurrency of the two works was handled in the most careful way we could.
> >
> > Regarding private response: We made some responses private because we did not think that discussing the reviewing (and decision) process of another conference would be appropriate for a public forum, and, as R2 notes, these would not be available to a reviewer in a double-blind review process.

---

> > > ### Comment · AnonReviewer3 · 2019-11-15
> > > **Re: Private response**
> > >
> > > I see. I presumed that you set the response to private by mistake. I personally don't think there is anything wrong with revealing this information to the public, but I will discuss it with the area chair during the post rebuttal discussion period and if he/she agrees that it's better to not make this information public, I will edit my comment.

---

### Official Review · AnonReviewer2 · 2019-10-23
**Official Blind Review #2**

**Rating:** 3

**Review:**

===== Summary =====
The paper presents a study about the representations learned by neural networks trained using robust optimization — a type of optimization that requires the model to be robust to small perturbations in the data. Specifically, the paper presents results of ResNet-50 trained on ImageNet with standard optimization and robust optimization. The paper draws three main insights from studying the learned representations of the standard and robust networks. First, the representation of the robust network is approximately invertible. In other words, when recovering an image by matching the representation of a random image to the representation of a target image by adding noise, the recovered images are semantically similar to the target image; the recovered images look similar to a human. Moreover, this is also demonstrated with images from outside of the distribution of the training data. Second, the representation of the robust network, unlike the representation of standard network, shows semantically meaningful high level features without any preprocessing or regularization. This leads to the final insight, feature manipulation is easier in robust networks. This is demonstrated by adding noise to an initial image in order to maximize the activation of a specific higher level feature and stopping early to preserve most of the other features of the original image.

Contributions:
1. The paper demonstrates that robust optimization enforces a prior on the representation learned by neural networks that results in high correspondence between the high-level features of an image and its representation in the network, i.e., similar images share similar representations.
2. The paper shows that the features learned by networks trained using robust optimization are semantically meaningful to humans without having to use any form of preprocessing.
3. The paper demonstrates that robust networks facilitate feature manipulation by injecting noise that maximally activates one of the features in the representation.

===== Decision =====
I consider that this paper should be accepted. The paper does not introduce any new algorithm or shows any theoretical results, but it is a great source of insight and intuition about robust optimization and deep learning. Moreover, the paper excels at the presentation and careful study of each of the main findings and it is well framed within the robust optimization literature.

===== Comments and Questions =====

There is still a major question that the paper does not directly address, but that is very relevant to robust optimization. Given that robust optimization seems to result in better-behaved and semantically meaningful representations, as evidenced by the findings in the paper, why is it that the performance of the resulting networks, in terms of classification accuracy, is lower than the performance of standard networks (trained with standard optimization)? It seems counter-intuitive that the robust network have worse accuracy than the standard network  given that it is more robust to small perturbations. I am curious if we could obtain any insights about this issue based on what has already being done in the paper. For example, are there any salient features in the images that the standard network classifies correctly but the robust network does not?

=== Minor Comments ===
1. I think the operator in Equations (1) and (4) should argmin since the noise is being added to x_1 in order to obtain x’_1.

2. What is the meaning of the error bars in Figure 4? I think this should be mentioned in the caption.


**Experience Assessment:**

I have read many papers in this area.

**Review Assessment: Checking Correctness Of Derivations And Theory:**

I carefully checked the derivations and theory.

**Review Assessment: Checking Correctness Of Experiments:**

I carefully checked the experiments.

**Review Assessment: Thoroughness In Paper Reading:**

I read the paper thoroughly.

---

> ### Author Response · Authors · 2019-11-11
> **Response #2**
>
> Thank you for your comments and suggestions about our work.
>
> As for the question about the robust models being less accurate, we agree that this is an interesting direction of study. Indeed, this question has been the focus of many recent papers in adversarial robustness (e.g., Su et al. 2019, Tsipras et al. 2019, referenced in our manuscript). Of these, Tsipras et al. (2019) provides a theoretical model very similar to what the reviewer is suggesting: that there are features that are predictive but not robust (and hence not human-meaningful). This view might also provide some insight into why robust models seem to have better feature representations as we observe in our work (i.e., robustness prevents the model from learning these brittle features).
>
> Comments (fixed in the revision):
> - Thank you for pointing out the typo in (1) and (4), we have corrected this.
> - The error bars are over random draws of the source and target images, we have modified the figure caption to reflect this.

---

> > ### Comment · AnonReviewer2 · 2019-11-15
> > **Updated rating**
> >
> > Thank you for your reply and clarifications. I don't want to take you into roller coaster ride throughout this update so I'll be upfront in saying that I have change my rating to a weak reject.
> >
> > I have been following the conversation between the authors and reviewer #3 and I also read the anonymized version of the Neurips paper that anonymous reviewer #3 kindly provided. I've tried to ignore the discussion about the chain of dependencies between both papers and about which paper was uploaded to arXiv first because that information wouldn't be available to us if this was a fully anonymized process. However, the bottom line is how novel the idea introduced by this paper is given the existing literature.
> >
> > I agree with reviewer #3 about how both papers deal with the same phenomenon: adversarially robust networks learn features that have high correspondence with the high-level features in natural images, which are semantically meaningful to humans. However, I also agree with the authors' statement about the first paper (Neurips paper) being a downstream application of this phenomenon while the current paper is a more in depth study of the phenomenon. In fact, the paper excels at this latter point since its presentation of the phenomenon is very well written and intuitive. Nevertheless, the applications presented in the paper are not as novel as the paper suggests. For example, the feature manipulations done in Section 4.2.1 of these paper are very similar to the feature paintings presented in Section 3.5 of the Neurips paper. It is true that the Neurips paper references this paper as a source of inspiration. However, since this paper would be published after the Neurips paper, it should not present the idea of manipulating features in adversarially robust networks as completely novel. Another example is the relationship between representation inversion and image generation that reviewer #3 highlighted in their updated respond.
> >
> > I want to emphasize that I like the argument that the paper is trying to make and how it is trying to dig deeper into this phenomenon. Moreover, I think the quality of the presentation and writing are excellent. However, the claims about the novelty of the application of this phenomenon are not well justified given that the Neurips paper has already being accepted and will be published in the proceedings of that conference in December.
> >
> > The question that stands is what could be changed to the paper to make it a stronger submission to future conferences? What I found most valuable about this paper was the study of the phenomenon itself and not necessarily the applications. Thus, one possibility would be, as suggested by reviewer #3, to study the representations of the networks trained using other adversarial optimization methods. Another very interesting alternative would be, as the authors suggested, to study the type of features that are predictive but not necessarily robust, which could guide the design of methods that could achieve both high accuracy and robustness to adversarial attacks.

---

> > > ### Author Response · Authors · 2019-11-15
> > > **Re: Updated rating**
> > >
> > > Thanks for your comments. As we point out to R3 somewhat deep in the thread (and as you acknowledge in your comment),  the papers do not in fact cover the same phenomenon, the NeurIPS submission shows that downstream applications are possible with robust models, while this paper is about representation learning.
> > >
> > > There is also a key factual inaccuracy w.r.t. inversion vs. generation: we would refer the reviewer to our reply to R3, where we clarify this---generation and inversion are actually two *completely* different tasks (crucially, generation doesn't say anything about representation learning, whereas there have been many papers studying *solely* inversion in the context of representation learning).
> > >
> > > Please let us know if we can make any further clarifications. The only real similarity between the two works is in the feature painting vs feature manipulation. We emphasize that the latter was the inspiration for the former as cited in the camera-ready paper distributed, which is why it is claimed novelly here. (For the purposes of the review process though, given the confusion, we are fine with the reviewers considering feature painting prior work---in our view, our work still provides novelty in this case both in thoroughly studying feature visualization through the lens of representation learning, and studying the inversion problem, which is completely orthogonal to the prior paper.)

---

### Author Response · Authors · 2019-11-11
**Revision and responses uploaded**

We thank all the reviewers for their thoughtful comments and suggestions regarding our work. We have updated (a) the manuscript to fix the notational typos in Equations (1), (4), and (5) pointed out by the reviewers, as well as some minor wording/grammar/formatting edits; and (b) responded inline to each review.

---

### Decision · Program_Chairs · 2019-12-19

**Decision:**

Reject

**Comment:**

The paper proposes recasting robust optimization as regularizer for learning representations by neural networks, resulting e.g. in more semantically meaningful representations.

The reviewers found that the claimed contributions were well supported by the experimental evidence. The reviewers noted a few minor points regarding clarity that seem to have been addressed. The problems addressed are very relevant to the ICLR community (representation learning and adversarial robustness).

However, the reviewers were not convinced by the novelty of the paper. A big part of the discussion focused on prior work by the authors that is to be published at NeurIPS. This paper was not referenced in the manuscript but does reduce the novelty of the present submission. In contrast to the current submission, that paper focuses on manipulating the learned manipulations to solve image generation tasks, whereas the current paper focuses on the underlying properties of the representation. Since the underlying phenomenon had been described in the earlier paper and the current submission does not introduce a new approach / algorithm, the paper was deemed to lack the novelty for acceptance to ICLR.